# BETTERBOOST - INFERENCE OF GENE REGULATORY NETWORKS WITH PERTURBATION DATA

**Achille Nazaret & Justin Hong**
Department of Computer Science
Columbia University
New York, USA
`{achille.nazaret,justin.hong}@columbia.edu`

## ABSTRACT

The introduction of large-scale, genome-wide, single-cell perturbation datasets provides the chance to learn a full gene regulatory network in the relevant cell line. However, existing gene regulatory network inference methods either fail to scale or do not explicitly leverage the interventional nature of this data. In this work, we propose an algorithm that builds upon GRNBoost by adding an additional step that complements its performance in the presence of labeled, single-gene interventional data. Applying BetterBoost to the CausalBench Challenge, we demonstrate its superiority over the baseline methods in inferring gene regulatory networks from large-scale single-cell perturbation datasets. Notably, BetterBoost exhibits significantly improved performance when non-zero fractions of labeled interventions are available, highlighting the effectiveness of our approach in leveraging interventional data for accurate gene regulatory network inference.

## 1 INTRODUCTION

The introduction of large-scale, genome-wide, single-cell perturbation datasets (Replogle et al., 2022; Dixit et al., 2016) provides a valuable opportunity to learn comprehensive gene regulatory networks. However, existing methods for gene regulatory network inference fail to scale (Brouillard et al., 2020; Sethuraman et al., 2023) or lack explicit utilization of the interventional nature of this data (Moerman et al., 2019; Passemiers et al., 2022). Methods that fail to scale often have algorithmic complexity issues, such as those encountered when computing the exponential of large matrices. On the other hand, methods capable of handling datasets with over 10,000 genes (Moerman et al., 2019; Passemiers et al., 2022) often treat the data as observational, thereby overlooking the valuable interventional information. While incorporating interventional data can enhance the predictive power of models that treat the data as observational, these models fail to fully exploit causal inference principles that aid in identifying causal relationships. To address these challenges and facilitate the advancement of causal inference methods on single-cell data, the CausalBench framework has been developed (Chevalley et al., 2022), and the CausalBench challenge was organized within the ICLR 2023 Workshop on Machine Learning for Drug Discovery. In this paper, we introduce BetterBoost, our winning method for the CausalBench challenge.

BetterBoost builds on the baselines proposed in the CausalBench framework. Among the scalable models that do not incorporate interventional data, we found that GRNBoost (Moerman et al., 2019) performed the best. GRNBoost defines the target gene's parents as the target's most predictive genes using a prediction importance score $G_{i,j}$ from gene $i$ to gene $j$. We adapted the GRNBoost score $G_{i,j}$ into a score $B_{i,j}$ in our proposed method, BetterBoost, which leverages interventional data in complement to observational data. The score $B_{i,j}$ reduces to $G_{i,j}$ when only observational data is available and improves as more interventional data becomes available.

BetterBoost assumes that if the dataset was generated by a causal model, the observed data's joint distribution can be factorized as:

$$p(\boldsymbol{x}_1 \dots \boldsymbol{x}_G) = \prod_{i=1}^{G} p(\boldsymbol{x}_i | \mathrm{Pa}(\boldsymbol{x}_i)). \tag{1}$$

If a candidate gene is a parent of the target, it will be a good predictor for the target, as GRNBoost assumes. But with labeled, interventional data, one can attempt to identify the true causal parents of a given observed variable $\boldsymbol{x}_i$ by looking at the effects of interventions on the candidate parents of $\boldsymbol{x}_i$. In particular, in a sample where a candidate parent gene is knocked down, the perturbed gene will only remain a good predictor for the target gene if it is a true causal parent of the target. Hence, if knocking down a candidate gene leads to a statistically significant prediction of the target gene, it indicates strong evidence of a causal relationship directed from the candidate parent to the target gene. We leverage the impact of knocking down candidate genes in the prediction importance score of BetterBoost.

We find that BetterBoost performs significantly better than leading methods GRNBoost (Passemiers et al., 2022) and DCDI (Brouillard et al., 2020) on provided sample data according to the challenge metric, average Wasserstein distance. Below, we detail the proposed method and go over the preliminary results of BetterBoost and relevant baselines on sample datasets.

## 2 METHODS

In this section, we restate the objective of the challenge and detail the algorithm, BetterBoost.

### 2.1 OBJECTIVE

The considered single-cell perturbational datasets each consist of a matrix of UMI counts per cell, $\boldsymbol{X} \in \mathbb{Z}^{+^{N \times G}}$, and associated interventional labels, $\boldsymbol{s} \in \{\text{unperturbed, unlabeled}, 1, \ldots, G\}^N$, for each cell. Note the interventions can only affect at most one gene, which can be achieved via high-precision CRISPRi technology (Larson et al., 2013). We denote the fraction of genes $g \in [G]$ with labeled interventional data as $\rho$.

Since ground truth causal network data does not exist for these datasets, a proposed causal graph is evaluated by the average Wasserstein distance which is defined as follows: for each edge in the inferred causal graph $(i, j) \in \hat{\mathcal{G}}$, the Wasserstein distance is computed between the distribution of $X_j$ in the unperturbed data and in the subset of data where $X_i$ is perturbed. Therefore, the average Wasserstein distance can be written as:

$$d(\hat{\mathcal{G}}) := \frac{1}{|\hat{\mathcal{G}}|} \sum_{(i,j) \in \hat{\mathcal{G}}} W_1(p(\boldsymbol{x}_j | \boldsymbol{s} = \text{unperturbed}), p(\boldsymbol{x}_j | \boldsymbol{s} = i)) \tag{2}$$

where $W_1$ denotes the first Wasserstein distance between two distributions.

The space of valid causal graphs, $\hat{\mathcal{G}}$ is constrained to $\{\hat{\mathcal{G}} : |\hat{\mathcal{G}}| \geq 1000\}$, but otherwise can include cycles and disconnected components.

### 2.2 ALGORITHM

We found GRNBoost to work the best in the observational case, i.e. no labeled interventional data, but fail to improve on this metric after adding strictly more information in the form of intervention labels. Thus, we developed a simple procedure for leveraging any available intervention labels. As previously mentioned, we assume that the true causal graph, $\mathcal{G}$ is a directed, acyclic graph (DAG), and therefore the joint distribution factorizes as in Equation 1. To identify if gene $j \in [G]$ is a strong candidate parent gene for a given target gene $i \in [G]$, we look if $j$ is predictive of the target gene $i$ in the dataset formed by observational data and the interventional data on gene $j$. For a true causal parent, we expect that when $j$ is knocked down, there will be a statistically significant shift in the distribution of observed UMIs of gene $i$ between observational and interventional data. Since we held no priors on the nature of causal effects, we chose to use the Kolmogorov-Smirnov (KS) test (Massey, 1951) to test these distributional shifts between observational and interventional data. Additionally, we used the Benjamini-Hochberg procedure to correct the p-values for multiple testing (Benjamini & Hochberg, 1995).

To formulate the new score used by BetterBoost to rank the impact of gene $i$ on gene $j$, we write $G_{i,j}$ the predictive score of gene $i$ on gene $j$ computed by GRNBoost, and $p_{i,j}$ the Benjamini-Hochberg corrected KS test p-value of the impact of knocking down gene $i$ on gene $j$. If no interventional

Table 1: Average Wasserstein Distance of Methods on RPE1 Perturb-seq dataset

| Method | $\rho = 0$ | $\rho = 0.25$ | $\rho = 0.5$ | $\rho = 0.75$ | $\rho = 1.0$ |
|---|---|---|---|---|---|
| DCDI | 0.126 | 0.126 | 0.127 | 0.125 | 0.130 |
| GRNBoost | 0.115 | 0.106 | 0.106 | 0.106 | 0.106 |
| GRNBoost-1000 | **0.151** | 0.147 | 0.146 | 0.146 | 0.145 |
| BetterBoost | **0.151** | **0.398** | **0.531** | **0.599** | **0.636** |

data was available on $i$, we set all p-values $p_{i,*}$ to 0.05, as to neither strongly accept nor reject hypotheses for these interactions. We then define the score $B_{i,j} = (-p_{i,j}, G_{i,j})$ that we sort from larger to smaller (in lexicographic order).

For some desired number of edges, $K$, BetterBoost returns the $K_B := \min(K, |\{(i,j) : B_{i,j}[0] \geq -0.05\}|)$ candidate edges with the smallest $H$ score and acceptable p-values. The $K_B$ candidate edges will have the smallest p-values for the KS test up to 0.05, which can include gene pairs where no interventional data and hence no p-value was available. Since the p-values of these gene pairs were set to 0.05, this ranking will favor in practice the edges of pairs with small p-values (obtained from combined interventional and observation data) followed by the edges with the highest GRNBoost scores $G_{i,j}$ (from observational data only). Typically, this results in more of the final edges being chosen by p-value than by GRNBoost score as more labeled interventional data becomes available.

## 3 RESULTS

We compared BetterBoost to the two suggested baseline methods, GRNBoost and DCDI, on the RPE1 perturbational data from (Replogle et al., 2022). The methods were evaluated with varying fractions of available labeled interventional data, ranging from 0.25 to 1.0. In order to comply with the challenge requirements, we choose to return $K = 1000$ edges for the challenge. By default, GRNBoost returns all edges with non-zero importance, so we additionally tested against a variant of GRNBoost that only returns the 1000 top importance edges.

We found that for every fraction of labeled interventional data, $\rho$ considered, BetterBoost improved significantly on the average Wasserstein metric. Additionally, we found that the improvement in the metric correlated perfectly with $\rho$ as shown in Table 1.

*Remark: We haven't tuned DCDI; the reported results are from running the provided baseline.*

## 4 DISCUSSION

Our proposed method, BetterBoost, utilizes labeled interventional data to identify the true causal parents of a given observed variable by looking at the effects of interventions on candidate parents. BetterBoost significantly outperforms leading methods GRNBoost and DCDI on provided sample data according to the challenge metric, average Wasserstein distance. In conclusion, our results suggest that BetterBoost is a promising gene regulatory network inference method.

BetterBoost can be extended for future work to consider the invariance property of causal relationships mentioned previously. Currently, if a chain of strong causal effects exists, $x_i \rightarrow x_j \rightarrow x_k$, BetterBoost will likely assign an edge from $x_i \rightarrow x_k$. However, if the interventional data on $x_j$ is present and labeled, one can identify that an edge does not exist between $x_i$ and $x_k$. This scenario also exposes a shortcoming of the average Wasserstein metric, which would not penalize the presence of such an edge in the inferred graph.

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
