# OpenReview forum: "BetterBoost - Inference of Gene Regulatory Networks with Perturbation Data"
_GSK.ai/2023/CBC_

### Official Review · Reviewer_f61T · 2023-04-26

**Rating:** 8
**Confidence:** 5

**Review:**

The authors propose an updated version of GRNBoost, that takes into account the available interventional data. To do so, they compute statistical test of differential expression on the available intervened gene, and then used the p value along with the GRNBoost score to rank the edges. In practice, the final edges are mainly chosen by p-values, and if not enough edges can be chosen that way, the GRNBoost score is used. One thing that is unclear is whether edges with a GRNBoost score of 0 can still end up being chosen through the p-value. It is also unclear how edges would be chosen outside of taking the top 1000, as was the optimal solution for this challenge. Furthermore, a p-value higher than 5% does not necessarily mean that there is no interaction, which in practice could mean that some true interactions are left out.

In conclusion, I think that the proposed solution is a very nice attempt at including the interventional information along with GRNBoost. I think an interesting avenue of future work would be to find a way of computing a unique score value instead of a pair.

Side-note: please kindly consider citing the causalbench paper in your report!